# Automatic light-adjusting electrochromic device powered by perovskite solar cell

Huan Ling [1], Jianchang Wu [1], Fengyu Su [1,2,3✉], Yanqing Tian [1✉] & Yan Jun Liu [3✉]

Electrochromic devices can modulate their light absorption under a small driving voltage, but the requirement for external electrical supplies causes response-lag. To address this problem, self-powered electrochromic devices have been studied recently. However, insensitivity to the surrounding light and unsatisfactory stability of electrochromic devices have hindered their critical applications. Herein, novel perovskite solar cell-powered all-in-one gel electrochromic devices have been assembled and studied in order to achieve automatic light adjustment. Two alkynyl-containing viologen derivatives are synthesized as electrochromic materials, the devices with very high stability (up to 70000 cycles) serves as the energy storage and smart window, while the perovskite solar cell with power-conversion-efficiency up to 18.3% serves as the light detector and power harvester. The combined devices can automatically switch between bleached and colored state to adjust light absorption with variable surrounding light intensity in real-time swiftly, which establish significant potentials for applications as modern all-day intelligent windows.

[1] Department of Materials Science and Engineering, Southern University of Science and Technology, Shenzhen, China. [2] Academy for Advanced Interdisciplinary Studies, Southern University of Science and Technology, Shenzhen, China. [3] Department of Electrical and Electronic Engineering, Southern University of Science and Technology, Shenzhen, China. ✉email: fysu@sustech.edu.cn; tianyq@sustech.edu.cn; yjliu@sustech.edu.cn

ncreased cooling demands in modern buildings and automobiles have led to increase in the consumption of energy[1–3]. Thus, it is urgent and meaningful to develop new technologies to efficiently modulate the light–heat flow between residences and surrounding environments. Smart electrochromic (EC) windows offer a promising approach since they respond to potential bias as transmittance changes smartly[4–7]. Though remarkable energy savings were achieved, the dependence on external power caused response-lag in light modulation for electrochromic devices (ECDs). In recent years, to address this issue, photovoltaic (PV) cell-powered ECDs have attracted intensive attention[8,9]. For examples, Dyer et al.[10] presented solar-powered transparent-magenta EC window with vertically integrated configuration containing two PV cells and a polymer ECD. Cannavale et al.[11] reported a combined semi-transparent PV and WO₃-based ECD which was able to adjust the multifunctional device from a neutral semi-transparent to dark blue. Xia et al.[12] demonstrated that EC batteries were charged by perovskite solar cell (PSC) accompanied by color changes from transparent to blue color, with reduced graphene (rGO)-connected bilayer NiO nanoflake as the cathode and WO₃ nanowire as the anode. These investigations dominantly utilized a sandwich configuration with a long-time coloration reservation; however, the ECD with layer-by-layer structure could easily turn to a colored state under strong light but failed to convert to transparent immediately when sunlight became weak only if a reverse potential bias was applied. In other words, conventional self-powered ECDs could not detect the change of sunlight intensity sensitively, especially to weakening light, which made the balance of outdoor vision and light–heat adjusting difficult. Additionally, the low PCE of PV cell, long response time, and unsatisfactory stability of ECD have limited these critical applications[13].

Among the configuration of ECDs, the all-in-one gel-type ECD refers to a device which is composed of an EC material, complementary material, and electrolyte in a homogeneous gel[14]. Unlike the conventional ECD with layer-by-layer sandwich configuration, all-in-one ECDs have a simpler structure and can convert to a transparent state without reversed voltage in a short duration[15]. Viologens, also known as bipyridine salts, have been explored intensively in all-in-one gel type ECDs[16] and energy storage devices[17–19] owing to their superior electrochemical properties in recent years. But the radical cations of viologens aggregate irreversibly on the electrode to form dimers, which would prevent the bleaching process[14]. These aggregations will eventually lead to poor stability.

Aiming to develop long-term stable, dynamic tunable, and multi-colors smart ECDs, two viologens including a novel mono-alkynyl-substituted viologen of 1-(pent-4-yn-1-yl)-[4,4′-bipyridin]-1-ium chloride (MPV) and a reported di-alkynyl-substituted viologen of 1,1′-di(pent-4-yn-1-yl)-[4,4′-bipyridine]-1,1′-diium dichloride (DPV) were synthesized and used for the fabrication of ECDs. The synthetic route of the two viologen derivatives was shown in Fig. 1. Introduced alkynyl groups would establish donor–acceptor interactions with electron-deficient species, including even radicals inspired by the famous Sonogashira coupling reaction[20] and a recently reported visible-light-induced radical alkynyl group migration reaction[21]. Herein, as a proof of concept, the gel-type ECD with the all-in-one configuration was integrated with PSC for automatic light-adjustment. PSC was selected as the energy harvester and light-detector due to its high photoelectric conversion efficiency, low cost, and mature assembly techniques[22–30]. The working schematic diagram of the combined devices was demonstrated in Fig. 2. PSC-powered ECDs can alter their colors automatically in real time depending on the surrounding solar intensity, and charged ECDs can serve as visual energy-statue ECSs (electrochromic supercapacitors) to supply other energy-consumption displays. To our delight, a large contrast ratio, excellent light sensitivity, and outstanding stability were obtained. This work may pave a new approach to develop dynamic light-tuning ECD-based commercial smart windows.

**Fig. 1 Synthetic route of MPV and DPV.** One-step synthesis of mono- or di-substituted viologen compounds.

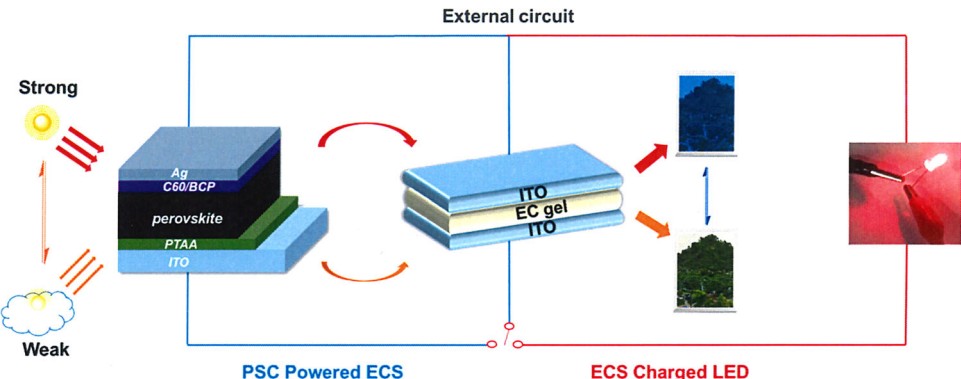

**Fig. 2 Schematic diagrams of device configurations and working principles of PSCs-powered ECS.** Perovskite solar cell (left) harvests solar energy to drive ECD/ECS (middle) to different colored states under different light intensities, and the stored energy of ECS (in deep colored state) can light a red LED (right).

## Results

**Performance of ECDs**. As shown in Fig. 3a, for DPV-based ECD, the first redox couple was detected at −0.57 V (anodic peak) and −1.01 V (cathodic peak), which could be attributed to the redox process of molecular dications of DPV$^{2+}$ and radical cation of DPV$^{+•}$. The second redox couple referred to the reaction of reversible transformation between DPV$^{+•}$ and full reduced species DPV$^{0}$. The detected anodic peak ($E_{pa}$) and cathodic peak ($E_{pc}$) were −1.24 V and −1.50 V, respectively. Nevertheless, the mono-substituent MPV demonstrated solo redox pair as shown in Fig. 3b, which pointed to the exchange of

MPV$^{•}$ and MPV$^{+}$. The $E_{pa}$(−1.00 V) and $E_{pc}$(−1.87 V) of MPV were lower than the values of first redox couple of DPV. The redox process was described in Eq. (1) and Eq. (2) for DPV and MPV, respectively. The detailed kinetic studies were given in Supplementary Note 1.

$$DPV^{2+} \underset{-e^-}{\overset{+e^-}{\rightleftarrows}} DPV^{+•} \underset{-e^-}{\overset{+e^-}{\rightleftarrows}} DPV^0 \quad (1)$$

$$MPV^+ \underset{-e^-}{\overset{+e^-}{\rightleftarrows}} MPV^• \quad (2)$$

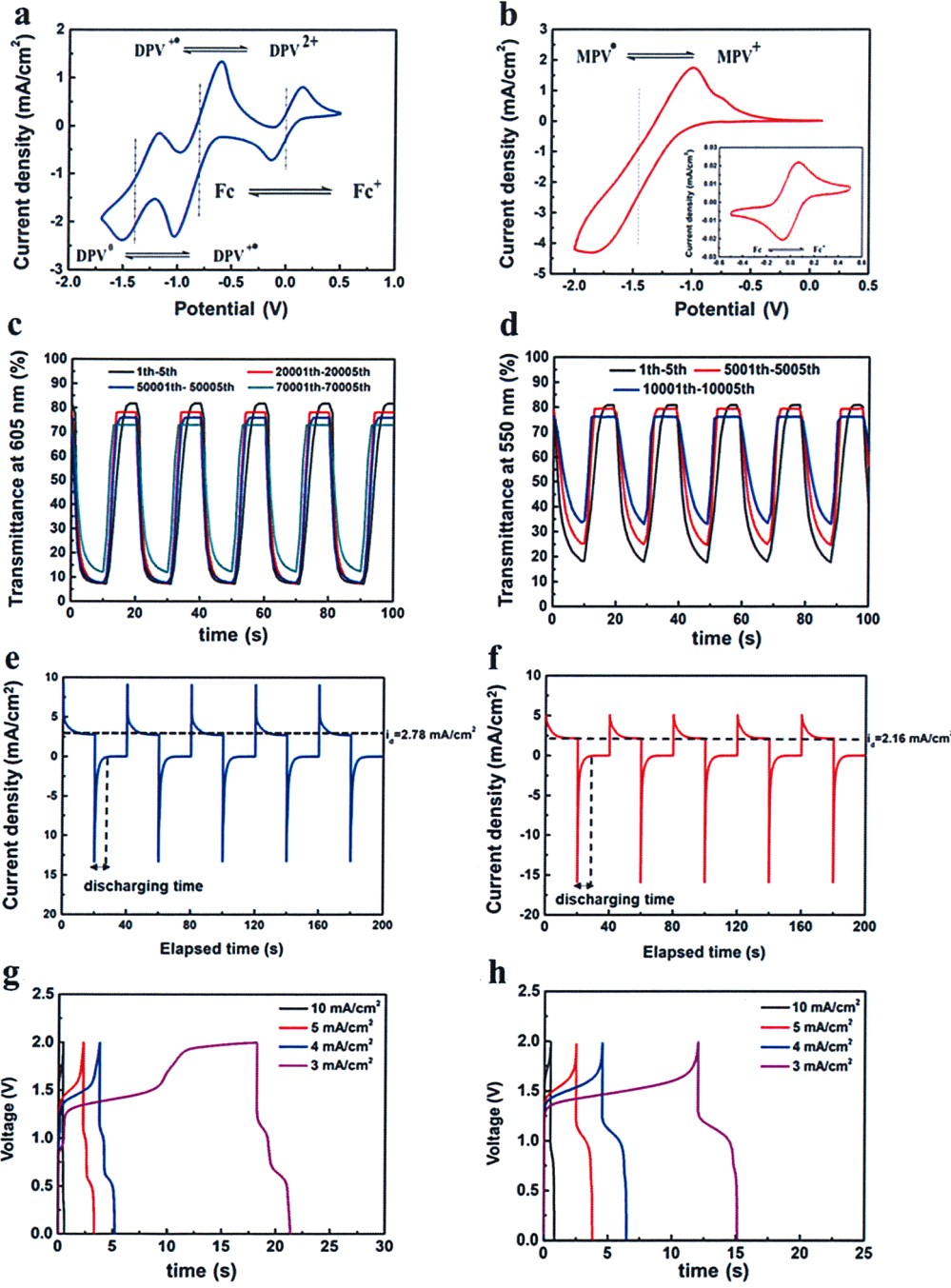

**Fig. 3 Performances of gel ECDs based on DPV and MPV.** (DPV: **a**, MPV: **b**) Cyclic voltammograms at a scan rate of 100 mV/s; (DPV: **c**, MPV: **d**) chronoabsorptometry at different switching cycles between colored state (1.6 V for 10 s) and bleached state (−0.3 V for 10 s); (DPV: **e**, MPV: **f**) current densities versus elapsed time switching between 1.6 V (colored state, 20 s) and 0 V (bleached state, 20 s); (DPV: **g**, MPV: **h**) galvanostatic charge–discharge (GCD) curves of ECSs under increasing current densities from 3 to 10 mA/cm$^2$.

One of the important parameters to evaluate ECD performance is coloration efficiency ($\eta$), which can be calculated from Eq. (3):

$$\eta = \Delta OD/\Delta Q = \log(T_b/T_c)/\Delta Q \qquad (3)$$

where $T_b$, $T_c$ were transmittance at a special wavelength in bleached and colored states, respectively; $\Delta Q$ was the required charge density for corresponding $\Delta OD$. $\eta$ was calculated by the slope of the linear region in the plots of $\Delta OD$–$\Delta Q$. Supplementary Fig. 3a and b gave the calculated values of the two viologen-based gel ECDs. The DPV exhibited higher $\eta$ of 89.4 cm²/C at 605 nm than that of MPV of 37.2 cm²/C at 550 nm, which indicated DPV can consume less charge to achieve the same $\Delta OD$. The CE value of 37.2 cm²/C for MPV was lower than the reported monoheptyl viologen-based ECDs (87.5 cm²/C at 546 nm by Oh et al.[31], 80.7 cm²/C at 552 nm by Kim et al.[32]).The value of 89.4 cm²/C of the DPV was higher than some symmetric di-substituent gel ECDs with heptyl viologen (87.3 cm²/C at 606 nm by Kim et al.[32]) and dinonyl viologens (36 cm²/C at 605 nm by Lu et al.[33]).

Stability was regarded as the most significant parameter for real-world applications. To investigate the long-term stability of the two alkynyl-based viologen gel ECDs, transmittance versus time was conducted with identical time intervals (10/10 s) and switching voltages (1.6/−0.3 V). The initial $\Delta T$ of DPV-based ECD as shown in Fig. 3c was 74.3% with the switching voltage of 1.6/−0.3 V. Only 3.2% and 6.1% of $\Delta T$ was lost after 20,000 and 50,000 cycles, respectively. It is noted that the transmittance at colored state only slightly increased from 7.3% to 7.7%, even after 50,000 cycles, which indicated DPV$^{+\bullet}$ did not aggregate on the electrode in the coloring process owing to the powerful interactions between alkynyl groups and radical cation when two alkynyl groups were introduced to one viologen molecule (Supplementary Fig. 2). Particularly, $\Delta T$ remained to be 60.9% after 70,000 cycles. To the best of our knowledge, the cycling stability of the DPV-based ECD was one of the best and even superior to most of the reported viologen-based ECDs (Supplementary Table 1), which demonstrated that the DPV possessed great potential for real applications. Compared to DPV, the cycling stability was relatively low for the MPV-based ECD (Fig. 3d). During the first five cycles, 62.6% of $\Delta T$ was obtained, and $\Delta T$ decreased to 43.2% after 10,000 cycles. The comparative low stability of MPV was ascribed to the aggregations of MPV$^\bullet$, which would impede the bleaching process. It is expected that the excellent stability of EC part would support the long-term work of the PSC-powered ECDs.

Supplementary Fig. 4a and b showed the dynamic changes of transmittance at 605 nm and 550 nm for DPV- and MPV-based gel ECDs under the switching voltages of 1.6 V and −0.3 V. The calculated response times for MPV are 5.6 s and 5.2 s to accomplish 90% of $\Delta T$ in a span between the bleached and colored states, which are longer than 2.4 s and 2.8 s for the DPV. The steady current density ($i_p$) in colored state was detected to be 2.78 mA/cm² for DPV and 2.16 mA/cm² for MPV at same applied voltage (Fig. 3e, f). These differences would be explained as following: the presence of radical dimers have hindered the generation of radical cation species. Instead, lower reduction potential was needed for DPV; additionally, the existence of two alkynyl groups in DPV not only restrained the formation of radical dimers, but also stabilized the produced radical cation species. As a result, DPV showed shorter coloring time and larger steady $i_p$ than those of MPV under the same switching voltages. On the other hand, MPV contained two oxidation reactions including MPV$^+$/MPV$^\bullet$ and MPV$^+$/(MPV$^\bullet$)$_2$ generated in the bleaching process, which led to longer bleaching time than that of DPV. In addition, the discharging time for ECDs based on DPV and MPV were 8.1 s and 9.1 s, which benefited to providing

electrical power for external energy consumption. And the fast coloring and self-bleaching time were extremely significant to the sensitivity of PSC-powered ECDs.

**Energy storage properties as ECSs.** Another fundamental function for the ECD was storing the charge from imported currents. The charging/discharging processes were accompanied with coloration and bleaching behaviors. Thus, it is likely to be regarded as a pseudocapacitor according to Fig. 3g, h. Galvanostatic charge–discharge (GCD) profiles with increasing current densities were recorded. The areal capacitor ($C$) was calculated from GCD curve utilizing Eq. (4):

$$C = I\Delta t/S\Delta V \qquad (4)$$

where $I$, $\Delta t$, $S$, $\Delta V$ represent discharging current, discharging time, active areal of capacitor, and potential window, respectively. For instance, the $C$ values were 6.7 mF/cm², 4.9 mF/cm², 5.0 mF/cm², and 2.2 mF/cm² at 3 mA/cm², 4 mA/cm², 5 mA/cm², and 10 mA/cm², respectively, for DPV-based ECS (Supplementary Fig. 5a). Supplementary Fig. 5b showed $C$ values of MPV-based ECS (7.1 mF/cm², 6.2 mF/cm², 5.5 mF/cm², and 3.5 mF/cm² at increasing current densities). It is observed that low $C$ values were obtained at high current densities and ECS based on MPV exhibited a higher $C$ value than DPV at same current density, which could be attributed to the faster diffusion process in ECS based on MPV. In addition, both ECSs demonstrated rapid charging/discharging process, especially very fast discharging; therefore, the coloring and bleaching processes would be completed in a short duration. Meanwhile, the stored energy in ECS could provide power for other electronics, which indicated the PSC-powered ECS has realized the functions of energy storage and reutilization.

**Performances of PSCs.** Reverse structure is widely used in PSCs, where the hole transporting layer is located in the light incident side. Usually, poly[bis(4-phenyl)(2,4,6-trimethylphenyl)amine] (PTAA) and C$_{60}$ were selected as hole transporting materials (HTMs) and electron transporting materials (ETLs) in inverted structures, while 2,2′,7,7′-Tetrakis[$N,N$-di(4-methoxyphenyl) amino]−9,9′-spiro-bifluorene (Spiro-OMeTAD) and TiO$_2$ are selected for traditional structure. However, the oxygen vacancy in TiO$_2$ (ref. [34]) and the dopant in Spiro-OMeTAD layer[35,36] would accelerate the degradation of perovskite. Therefore, devices with inverted structure were selected as the power for ECD[37,38]. Supplementary Fig. 6a demonstrated the exact structure of the PSC. The deposition method of PSCs was similar to our previous work[39]. The perovskite was deposited on HTM. PEDOT:PSS and PTAA are star HTMs for PSCs, device based on PEDOT: PSS always shows a low open-circuit voltage ($V_{oc}$) because of unmatched energy levels, which is harmful for powering the connected EC capacitor. Thus, aiming to achieve relatively high voltage to drive the ECD, we selected PTAA as the HTM. As depicted in Table 1, the PSC device can produce a $V_{oc}$ of 1.02 V with a short-circuit current of 22.8 mA/cm² and fill factor of 78.4% under illumination, and reasonable high PCE of 18.3% was achieved according to Supplementary Fig. 6b, which was better than reported work[40,41]. High PCE guaranteed the robust power

**Table 1 Parameters of fabricated PSCs.**

|  | PCE (%) | $V_{OC}$ (V) | $J_{SC}$ (mA/cm²) | FF (%) |
|---|---|---|---|---|
| Reserve scan | 17.7 | 1.0 | 22.1 | 80.2 |
| Forward scan | 18.3 | 1.02 | 22.8 | 78.4 |

PCE = $V_{oc} \times J_{sc} \times$ FF; $V_{oc}$: open-circuit voltage; $J_{sc}$: short-circuit current density; FF: fill factor.

for the ECDs. The PSCs generated the voltage and current, making the ECD change from transparent to a colored state, indicating the solar cell could power the ECD and the electrical power is stored in the ECD.

**Performances of PSC-powered ECDs**. To detect the light sensitivity of PSC-powered ECDs, we have investigated the alterations of UV-visible spectrum opposing different light density and monitor the color changes in natural sunlight. As shown in Fig. 4a, e, the whole transmittance in visible range increased gradually with the weakening of light density. Both ECDs turned to highly transparent automatically. The corresponding characteristic peaks for DPV- and MPV-based PSC-powered ECDs were 605 nm and 550 nm, which were consistent with spectrum under common extra voltage exhibited in Supplementary Fig. 7. Afterward, in natural climate, the ECDs demonstrated excellent self-powered, smartly tunable features. For the DPV-based ECDs shown in Fig. 4b, d, a deep blue color was detected under high light density since more excitons were generated in PSCs when higher solar energy from sunlight was harvested, which initiated larger current (or electrons) to reduce more $DPV^{2+}$ to $DPV^{+\bullet}$ species. With descending light density, the ECD demonstrated a light blue color because the current dropped. Finally, the ECDs transferred to highly transparent when the PSCs were blocked. Similarly, MPV-based ECDs were able to change the appearance depending on the natural surrounding sunlight. However, the distinction was that magenta-blue colors appeared with the drop of light density (Fig. 4f, h). The multi-colors of MPV-based ECDs were more convenient for us to evaluate the surrounding sunlight

intensity and the stored power quantity in variable weather according to the color type and depth. In addition, we also conducted the switching stability of PSC-powered ECD in strong light/no light surrounding under moisture conditions as shown in Fig. 4i, j, both PSCs-powered ECDs did not show any decay after 2000 s constant working ($\Delta T = 71\%$, 62% for DPV and MPV, respectively) in air condition, which revealed good switching stability and fast response time (<30 s) for both of these combined devices (Supplementary Fig. 8). In addition, both charged ECDs with deep coloration (under strong light intensity) could drive a red LED successfully (Fig. 4k, l) as supercapacitors. These brilliant results suggested that PSC-powered ECDs possess a remarkable advantage to modulate the sunlight and could serve as an intelligent-curtain. When exterior sunlight intensity changed frequently, the unpredictable optical radiation and energy flow problems lead to a poor living experience. Different from the present solutions like using air-conditions to cool down and curtains to shade individually, this intelligent-curtain addressed the two main problems together. As depicted in Fig. 5, with the increasing of light intensity from 6:00 to 12:00, the transmittance dropped and the appearance of ECD transferred to colored state gradually. Afterward, with the decline of light density in afternoon, the color of intelligent-curtain will be shoaled automatically to let more light in to warm the house. Finally, it gets highly transparent near the evening, which gives us a good insight to observe the scenery outside as a classic window. The annual energy consumption of the PSC-powered ECD is given in detail in Supplementary Note 2.

Colorimetric system was utilized to qualitatively characterize the colors through three parameters including $L^*$, $a^*$, $b^*$, where

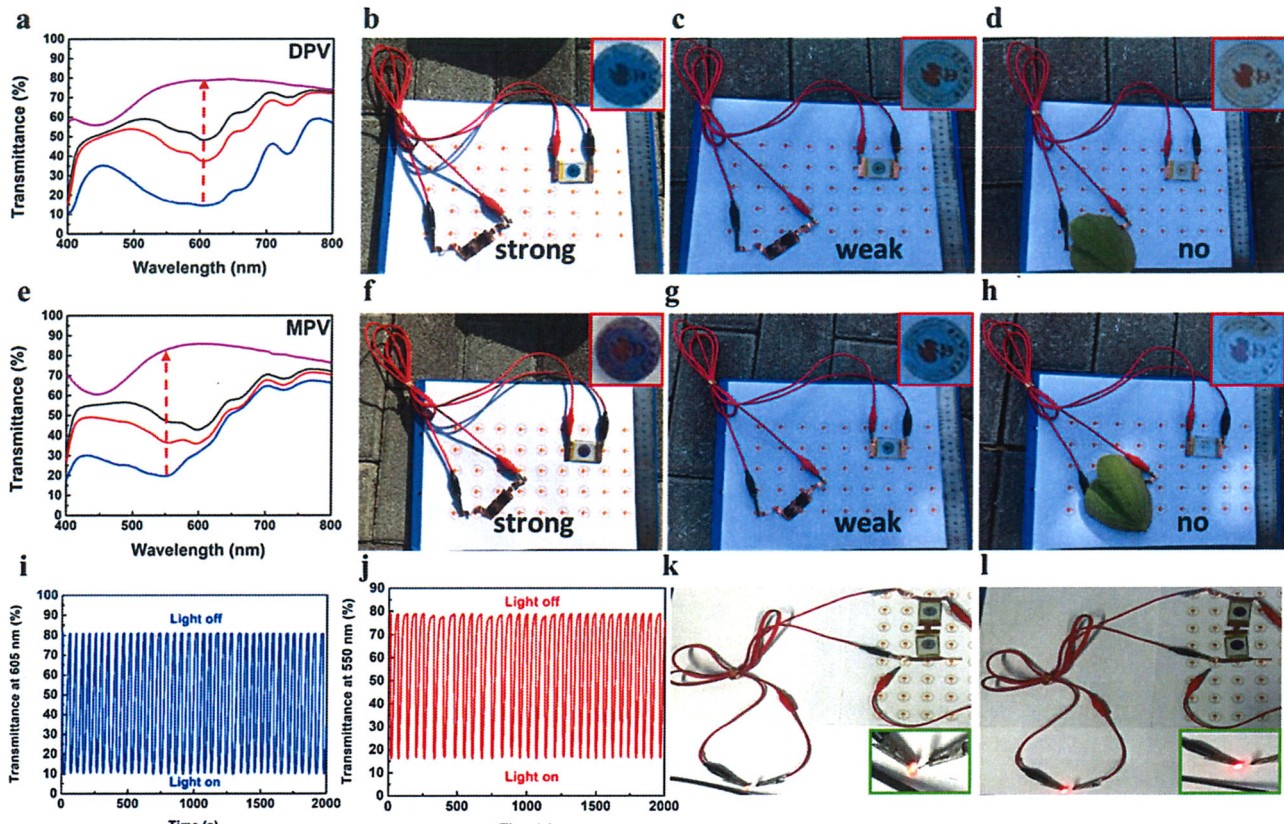

**Fig. 4 Performances of PSC-powered ECS based on DPV and MPV.** UV-visible spectra of ECDs powered by two connected PSCs with decreased light density for DPV (**a**) and MPV (**e**); PSCs powered ECD in natural surrounding with strong light, weak light, and no light (**b**, **c**, **d** for DPV-based ECD and **f**, **g**, **h** for MPV-based ECD); transmittance switching tests under strong light and no light of PSC-powered ECD at maximum absorption (**i**: DPV, **j**: MPV); lighting LED by two connected ECDs (**k** DPV, **l** MPV).

$L^*$ is lightness, and $a^*$ and $b^*$ were used to assess the green–red and blue–yellow colors, respectively. In order to evaluate the real-time response performances of intelligent ECDs, we simulate the change of outdoor light intensity through adjusting the distance between the illuminant and PSCs in situ as shown in Fig. 6a. When the distance $L$ was 0.05 m, the PSCs were exposed to the light with high intensity, resulting in the values of $L^*/a^*/b^*$ being 36.29/0.42/−34.55 (Table 2) and 40.83/17.19/−20.37 (Table 3) for DPV- and MPV-based ECDs, respectively, which indicated deep blue and purple colors were obtained. With the increasing of $L$, as well as the decreasing of light intensity, the $L^*$ value for both ECDs rises gradually, while the colors simultaneously became shallower, which were proved by the decreasing $|b^*|$ for DPV and decreasing $|a^*|$ and $|b^*|$ for MPV. When the illumination was removed, the high $L^*$ and low $|a^*|$ and $|b^*|$ suggested ECDs were converted to highly transparent. Figure 6b, c vividly demonstrated the real-time color changes under different light intensity. These results confirmed that the PSC-powered ECDs possess excellent light sensitivity to alter their transparency and colors according to the surrounding light intensity, which paved a new way to design unprecedented intelligent-curtain.

## Discussion

In summary, the ECDs based on dipentynyl substituted viologen, DPV, worked very well up to 70,000 cycles. The outstanding stability was attributed to the intermolecular interactions between viologen radical species and alkynyl groups. As a proof of concept, ECDs with all-in-one gel configuration were combined with PSCs to realize the automatically dynamic light-adjusting. Two self-powered and light-sensitive intelligent-curtains with MPV- and DPV-based devices as the ECD part, and PSCs as the energy harvester were successfully fabricated. The characterizations and tests were performed in indoor and outdoor circumstances. Among them, DPV-based ECD exhibited transparent-deep blue color as the light intensity changed, and MPV-based ECD demonstrated transparent-blue-magenta colors depending on the light intensity. High optical contrasts, excellent light sensitivity, and excellent stability were achieved. Furthermore, energy harvesting, storage, and light regulation could change real-time responding to the surrounding light sensitively. Smart modulation of the solar radiation contributes to visual comfort for people. When the sunlight is strong, PSC harvests more energy and drive the EC into dark state, EC could permit less solar radiation to enter the building, therefore people inside will feel more comfortable under a weaker light than outside; on the other hand, when the sunlight is weak, PSC harvests less energy and drive the EC into relative bright state, EC could permit more solar radiation to enter the building, thus people inside will feel more comfortable under a brighter light. In addition, we can evaluate the light density and stored power by the colors and shades of ECDs. And the stored electrical energy successfully provided power to a LED. These extraordinary results suggest huge potentials of PSC-powered ECDs being applied for all-day intelligent-curtains and advanced electronic displays. However, considering the latent toxicity of DPV and MPV, strict sealing work would be required if PSC-powered ECDs were applied in modern buildings and automobiles.

## Methods

**Materials and synthesis.** The commercially available raw chemicals were used without further purification. 4,4′-Bipyridine was obtained from J&K Chemical (Beijing, China). HCl was purchased from Damao Chemical Ltd. (Shanghai, China). 5-Chloro-1-pentylene was obtained from Aladdin (Shanghai China). All solvents were purchased from XiLong Scientific (Guangzhou, China). ITO glass was purchased from NSG Group (Osaka, Japan).

Nuclear magnetic resonance (NMR) spectra were acquired through NMR (Bruker 400M, Bremen, German). The electrochemical properties were characterized by electrochemical workstation (Ametek Parstat 3000A-DX, Berwyn, USA). The optical properties were measured by UV-vis spectrophotometer (PerkinElmer L6020365, Waltham, USA) and Colorimeter (CHN Spec CS-820, Guangzhou, China) combined with Waveform Generator (Rigol DG4102 Function/Arbitrary, Suzhou, China). A digital camera was used to record the photographs of the devices.

**Synthesis of 1-(pent-4-yn-1-yl)-[4,4′-bipyridin]-1-ium chloride (MPV).** A total of 0.3 g (1.9 mmol) of 4,4′-bipyridine and 0.176 g (1.7 mmol) of 5-chloro-1-pentylene were dissolved in 2 ml of DMF. The mixture was heated to 90 °C and stirred for 16 h. After that, the residue was cooled to r.t. and then concentrated under reduced pressure. Finally, the mixture was purified by fresh silica gel to give the 1-(pent-4-yn-1-yl)-[4,4′-

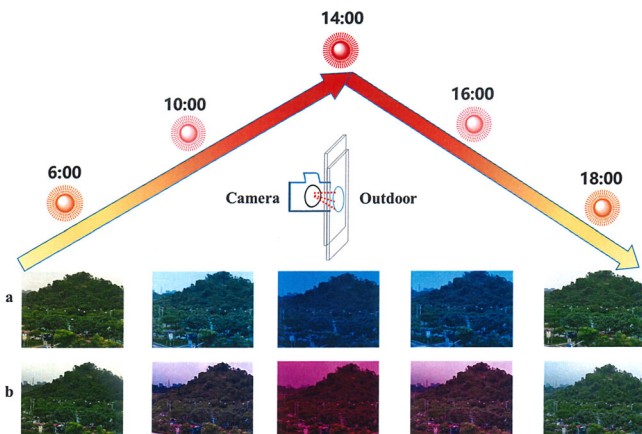

**Fig. 5 Automatic light adjustment of PSC-powered ECDs in real-time.** Digital images of outdoor sceneries viewed through PSC-powered ECDs by blocking device on the lens of the camera at different times (**a**: DPV; **b**: MPV).

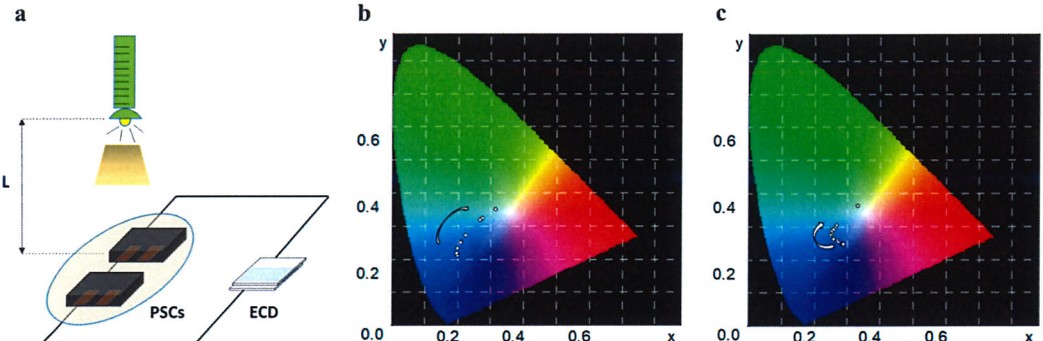

**Fig. 6 Colorimetric tests of PSC-powered ECDs under different light intensities. a** Schematic diagram for colorimetric test of ECD; colorimetric diagrams of DPV (**b**) and MPV (**c**) based PSC-powered ECDs with descending light intensity.

**Table 2 $L^{\star}$, $a^{\star}$, and $b^{\star}$ changes with different light density of DPV-based PSC-powered ECD.**

| L (m) | $L^{\star}$ | $a^{\star}$ | $b^{\star}$ |
|---|---|---|---|
| 0.05 | 36.29 | 0.42 | −34.55 |
| 0.1 | 36.16 | 0.57 | −34.47 |
| 0.2 | 39.13 | −3.07 | −34.87 |
| 0.3 | 43.61 | −8.26 | −32.72 |
| 0.4 | 47.59 | −11.6 | −29.45 |
| 0.5 | 48.47 | −12.18 | −28.62 |
| 0.6 | 52.62 | −14.03 | −24.3 |
| 0.7 | 64.42 | −13.25 | −9.84 |
| 0.8 | 66.21 | −12.44 | −7.42 |
| 1 | 75.09 | −6.99 | 4.66 |
| No light | 75.97 | −6.29 | 5.95 |

**Table 3 $L^{\star}$, $a^{\star}$, and $b^{\star}$ changes with different light density of MPV-based PSC-powered ECD.**

| L | $L^{\star}$ | $a^{\star}$ | $b^{\star}$ |
|---|---|---|---|
| 0.05 | 40.83 | 17.19 | −20.37 |
| 0.1 | 41.82 | 16.87 | −20.9 |
| 0.2 | 46.27 | 11.15 | −22.15 |
| 0.3 | 47.17 | 9.97 | −22.26 |
| 0.4 | 51.23 | 3.23 | −23 |
| 0.5 | 56.13 | −2.9 | −22.65 |
| 0.6 | 58.75 | −4.77 | −21.76 |
| 0.7 | 61.76 | −7.24 | −20.04 |
| 0.8 | 65.17 | −8.17 | −16.13 |
| 1 | 77.88 | −3.93 | 11.1 |
| No light | 78.82 | −3.66 | 11.82 |

bipyridin]-1-ium chloride (MPV) as a gray solid (0.1 g, 20% yield). $^{1}$H NMR (400 MHz, deuterium oxide) δ 9.09 (d, $J = 6.7$ Hz, 2H), 8.48 (d, $J = 6.3$ Hz 4H), 4.80 (t, $J = 6.8$ Hz, 4H), 2.39–2.05 (m, 10H). $^{13}$C NMR (101 MHz, deuterium oxide) δ 153.98, 150.01, 145.05, 142.51, 126.09, 122.47, 82.08, 70.78, 60.29, 28.77, 14.56. HRMS $m/z$ calcd for $C_{15}H_{15}N_2$ ([M]$^{+}$) 223.1230, found 223.1231.

**Synthesis of 1,1′-di(pent-4-yn-1-yl)-[4,4′-bipyridine]-1,1′-diium dichloride (DPV).** A total of 0.3 g (1.9 mmol) of 4,4′-bipyridine and 0.49 g (4.8 mmol) of 5-chloro-1-pentylene were dissolved in 2 ml of DMF. The mixture was heated to 110 °C and stirred for 48 h. After that, the residue was cooled to r.t. and the precipitate was filtered and washed by DMF (3 × 2 ml), then dried in vacuum to give 1,1′-di(pent-4-yn-1-yl)-[4,4′-bipyridine]-1,1′-diium dichloride (DPV) as a gray solid (450 mg, 66% yield). $^{1}$H NMR (400 MHz, deuterium oxide) δ 8.94 (d, $J = 6.8$ Hz, 2H), 8.66 (s, 2H), 8.33 (s, 2H), 7.82 (s, 2H), 4.74 (t, $J = 6.8$ Hz, 2H), 2.40–1.98 (m, 5H). $^{13}$C NMR (101 MHz, D$_2$O) δ 150.24, 145.77, 127.04, 82.39, 71.13, 60.85, 28.77, 14.56. HRMS $m/z$ calcd for $C_{20}H_{22}N_2$ ([M]$^{2+}$) 145.0886, found 145.0889.

*Fabrication of ECDs.* The preparation of MPV- and DPV-based EC gels was described as follows: 0.1 mmol of viologen, 150 mg of propylene carbonate (PC), 23 mg of ferrocene, 160 mg of lithium bis-(trifluoromethane)sulfonimide, and 300 mg of PVB (polyvinyl butyral) were dissolved in 1.5 ml of dried CH$_3$OH. The mixture was stirred under N$_2$ atmosphere for 3 h and was placed for a period of time without stirring for eliminating bubbles during the gel preparation.

The prepared gel was injected into liquid crystal cells. The distance of 70 μm between the two electrodes was controlled by parafilm. The effective area of ECDs appeared as a circle with a radius of 0.8 cm ($S = 2$ cm$^2$).

*Fabrication of PSCs.* ITO glass was cleaned by sequentially washing with deionized (DI) water, acetone, and isopropanol (IPA). Before using, the ITO was cleaned by ultraviolet ozone (UVO) for 20 min. Then, the substrate was spin coated with a thin layer of PTAA (2 mg/ml in chlorobenzene) at 5000 r.p.m./min for 25 s, and annealed at 100 °C for 10 min. The perovskite layer was deposited by a two-step sequential spin-coating method. Then, 60 μL of PbI$_2$ precursor (600 mg of PbI$_2$ dissolved in 1 ml of DMF:DMSO = 95:5 by volume) was spin coated on the substrate at 1500 r.p.m. Then, the mixture solution of formamidinium iodide (FAI): methylammonium bromide (MABr): methylamine chloride (MACl, 60 mg: 6 mg: 6 mg in 1 ml of isopropyl alcohol (IPA)) was spin-coated onto the PbI$_2$ film at

1300 r.p.m. for 30 s, and then a thermal annealing at 150 °C for 15 min in an ambient air condition (20–30% humidity) was conducted. Then, the samples were transferred to a vacuum chamber. C$_{60}$ (40 nm thick)/2, 9-dimethyl-4, 7-diphenyl-1, 10-phenanthroline, (BCP, 8 nm thick)/Ag (100 nm thick) were thermally evaporated in a vacuum chamber with the base pressure of <10$^{-4}$ Pa.

Photocurrent density–voltage ($J$–$V$) curves were measured under AM 1.5G one sun illumination (100 mW/cm$^2$) with a solar simulator (Enlitech SS-F7-3A) equipped with 300 W Xenon lamp and a Keithley 2400 source meter. The light intensity was adjusted by an NREL-calibrated Si solar cell. During measurement, the cell was covered with a mask with an aperture of 0.0576 cm$^2$. The $J$–$V$ curves were recorded from −0.2 to 1.2 V (forward scan) or from 1.2 to −0.2 V (reverse scan) with 0.01 V steps, integrating the signal for 10 ms after a 10-ms delay.

**Reporting summary.** Further information on experimental design is available in the Nature Research Reporting Summary linked to this paper.

## Data availability

The data that support the findings of this study are available from the corresponding authors on request.

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

## Acknowledgements

The authors would like to thank Shenzhen fundamental research programs (JCYJ20170817111349280, JCYJ20170412152922553, and JCYJ20180305180635082), and the start-up fund of SUSTech (Y01256114), the High-level university construction fund for SUSTech (G01256018), the National Key Research and Development Project funding from the Ministry of Science and Technology of China (grants no. 2016YFA0202400 and 2016YFA0202404), the Peacock Team Project funding from Shenzhen Science and Technology Innovation Committee (grant no. KQTD2015033110182370). This paper is dedicated to the 10th anniversary of Southern University of Science and Technology.

## Author contributions

H.L. designed and prepared the devices and performed the electrochromic tests. J.W. designed and carried out the tests of the perovskite solar cells. H.L. and F.S. conceived the idea and wrote the manuscript. Y.T. and Y.J.L. supervised the whole project and revised the manuscript. All authors discussed the results and commented on the manuscript.

## Competing interests

The authors declare no competing interests.
