## [Peer Review File · Nature Communications]

REVIEWER COMMENTS

Reviewer #1 (Remarks to the Author):

The manuscript "Automatic light-adjusting electrochromic device powered by perovskite solar cell" by Tian et al. describes an automatically light-intensity tuneable electrochromic device powered by perovskite solar cell. The colour in the electrochromic device is automatically switched under different light environments. There have been a lot of papers that incorporate self-power characteristic, colour change and energy storage properties in one integrated device, e.g., *Energy Environ. Sci.*, 2015,8, 1578-1584; *Mater. Horiz.*, 2016,3, 588-595.

The only novel thing in this work compared to the previous work is that the deep blue colour can be automatically switched to bleached state as the light intensity decreases; while in the previous work a reverse bias is required to bleach the device. It is a good performance for smart electrochromic windows. However, the novelty of this engineering increment is insufficient for publishing on *Nature Communications*. The authors state the electrochromic device has energy storage capabilities. However, from the discharging curve, the energy storage capability is very weak, and the capacitance is quite small. The faded colour in the device under a weak light illumination is a contradiction with energy storage capabilities. The "memory effect" of colour change is usually accompanied with ion storage and energy storage. Thus, it is not reasonable for the authors to claim the energy storage capabilities in the paper. Additionally, the solar cell and electrochromic device only show a comparable performance with the previous works, which have limited advances for a high-impact journal, e.g., *Nature communications*.

Reviewer #2 (Remarks to the Author):

This manuscript entitled "Automatic light-adjusting electrochromic device powered by perovskite solar cell" reported multi-functions of two ECDs based on synthesized alkynyl-containing viologen derivatives, DPV and MPV. Results are attractive that the absorption of the ECDs can be automatically modulated by the light intensity shined on the perovskite solar cells. This assembled PSCs-powered ECD system may have potential applications as smart windows. In addition, the ECDs can also store extra energy as the electrochromic supercapacitors to drive LEDs. The results are very interesting and promising, and may cause broad interests of researchers in relevant research areas. However, there are several non-accurate information, which need to be corrected and more accurate information needs to be added. I recommend this manuscript to be accepted for publication after minor revision.

1. The position of Fig.3 (c1) should be adjusted, since the maximum value of x-axis is supposed to be 2000 instead of 200.
2. Authors mentioned the ECDs showed an ultrahigh stability up to 70000 cycles. How did authors come to this statement comparable to other devices based on viologens? It is better to give a comparison table in the supplemental information.
3. What is the effect of the alkynyl group(s) on the performance of the ECD? Why is the di-pentynyl viologen-containing device much more stable than that of the mono-pentynyl viologen-containing one?
4. For the NMR, the J should be italic J.
5. In all the figures, the fonts should be united.
6. 1-(pent-4-yn-1-yl)-[4, 4'-bipyridin]-1-ium chloride (MPV) and a di-alkynyl-substituted viologen of 1, 1'-di(pent-4-yn-1-yl)-[4,4'-bipyridine]-1,1'-dium dichloride (DPV) were synthesized and used for the fabrication of ECDs. Scheme 1 should be indicated here, not in the method part.
7. GCD needs to be given after the first appearance of "Galvanostatic charge–discharge curves.." in the caption of Figure 2.
8. "The detailed kinetic studies were developed in Supporting information" should be "The detailed kinetic studies were given in Supporting information".
9. "To our delight, 18.3% of PCE was achieved according to Fig. S6b, which was better than reported work^{26, 27}. " this sentence needs reword. The PCE of 18.3% is only a good reasonable one, since many devices with high PECs were reported.
10. The discussion part is too short, I suggest authors move some discussions in the results part to this discussion section.

Reviewer #3 (Remarks to the Author):

The project of multifunctional devices is indeed an interesting topic, with particular interest in energy saving. The manuscript entitled "Automatic light-adjusting electrochromic device powered by perovskite solar cells", currently being under consideration for publication in this scientific journal, is very interesting and deals, at the same time, with: production of electricity by photovoltaic conversion, using perovskite-based solar cells, smart shielding of solar radiation using organic electrochromic materials (two types of viologens are used), storage of electrical energy, using the same viologens.

Nevertheless, in the face of such a significant commitment, revealed in the abstract, this work should undergo major revisions.

INTRODUCTION SECTION:

- it would be advisable to add a reference concerning the increase in global energy consumption;
- the authors should quote the fundamental works of S. Deb and C.G. Granqvist, about EC devices;
- the authors of the reference [5] are not correctly cited in the text;
- the authors argue that the use of electricity to activate electrochromic windows is a deeply felt problem in the scientific field: in truth, as reported by several authors, the annual energy consumption of a commercial smart window is about 2.5 kWh/m², compared to the achievable energy saving, which is at least an order of magnitude higher;
- the authors, on the other hand, do not refer to the benefits linked to visual comfort, connected to the modulation of the solar radiation obtainable;
- the authors declare that the so-called "memory effect" constitutes a limitation of common electrochromic glasses, but it is considered an asset for smart windows, worldwide;
- however Figure S8 shows that the self-bleaching of the viologens is rather rapid and does not guarantee the typical "memory effect" requirements, also cited by the literature of the field (the recommended values of memory effect range from 2h to 12h).
- With reference to the synthesis of viologens, the authors do not explain whether it was designed and implemented in this work for the first time or it is a protocol already appeared in literature;
- with reference to perovskite solar cells, some leading groups are not mentioned, which currently hold the records of photovoltaic conversion efficiency, also based on the annual NREL chart;

RESULTS SECTION:

The authors report a careful examination of the stability of the viologens, even if the fast self-bleaching would cause - probably - an increase in the number of cycles required for coloring/bleaching, during operation, compared to the typical battery-like devices, based on transition metal oxides; would the rapid bleaching of electrochromic systems would require a continuous bias to the device, to keep it dark, as in SPDs systems? The authors could estimate the annual energy consumption, if they intend to propose this device for integration in buildings;

- No reference is made to the stability of the perovskite-based solar cell, with inverted architecture, used in this device.
- Furthermore, it can be observed that this work announces the design and manufacture of a multifunction device, but only a juxtaposition of devices is presented in the text, mutually connected by an external electrical circuit.
- Other works of this type have appeared previously in the literature but they clearly declared to show just a "proof of concept", reporting at least a scheme of the device they intend to create. In fact, the difficulties of creating a multi-function device should not be overlooked, both in terms of layout and of processes: for example, what would the layout of the final device be? what is the relationship between the areas of the photovoltaic component and that of the electrochromic/capacitor component? is the photovoltaic component semi-transparent or opaque?
- Possibly, the authors should report some considerations about the potential toxicity of viologens used in this work.

Several typos are still visible in the current version of the manuscript: a thorough check would be useful.

October 27, 2020

Dear Reviewers:

Thank you for your comments and suggestions on our manuscript “Automatic light-adjusting electrochromic device powered by perovskite solar cell”. We have revised the manuscript carefully and made some changes and marked them in the revised manuscript. All of the responses to the comments and suggestions point by point were integrated in this response letter. Please see the details as following.

We expect that the current format will meet the requirements of “*Nature Communications*”.

Your favorable considerations will be greatly appreciated.

Thank you.

With best wishes.

Yanqing Tian, PhD., Professor
Materials Science and Engineering
Southern University of Science and Technology
Shenzhen, 518055, China
<http://www.researcherid.com/rid/C-1423-2013>

Reply to Reviewer 1

Comment. The manuscript “Automatic light-adjusting electrochromic device powered by perovskite solar cell” by Tian et al. describes an automatically light-intensity tuneable electrochromic device powered by perovskite solar cell. The colour in the electrochromic device is automatically switched under different light environments. There have been a lot of papers that incorporate self-power characteristic, colour change and energy storage properties in one integrated device, e.g., Energy Environ. Sci., 2015,8, 1578-1584; Mater. Horiz., 2016,3, 588-595.

The only novel thing in this work compared to the previous work is that the deep blue colour can be automatically switched to bleached state as the light intensity decreases; while in the previous work a reverse bias is required to bleach the device. It is a good performance for smart electrochromic windows. However, the novelty of this engineering increment is insufficient for publishing on Nature Communications. The authors state the electrochromic device has energy storage capabilities. However, from the discharging curve, the energy storage capability is very weak, and the capacitance is quite small. The faded colour in the device under a weak light illumination is a contradiction with energy storage capabilities. The “memory effect” of colour change is usually accompanied with ion storage and energy storage. Thus, it is not reasonable for the authors to claim the energy storage capabilities in the paper. Additionally, the solar cell and electrochromic device only show a comparable performance with the previous works, which have limited advances for a high-impact journal, e.g., Nature communications.

Answer: Thank you for pointing out your concern.

Firstly, the novel thing in this work was not only “the deep blue colour can be automatically switched to bleached state as the light intensity decreases”. We have found for the first time that alkynyl groups can stabilize the radical cation through p- π interactions, and hinder the direct stack of intramolecular radical species to prevent the undesired formation of quasi-reversible dimer in viologen based ECD (detailed discussion about this finding was given in supporting information of Fig S1 and Fig

S2 and first two paragraph). Through introducing two alkynyl groups, the all-in-one gel ECDs based on the simple di-pentynyl containing viologen exhibited very high stability (up to 70000 cycles), which demonstrated one of the best performance among the reported works so far. The intermolecular interactions between electron-donor groups and viologen radical species may provide a new approach to improve the stability of viologen-based ECD. Additionally, to the best of our knowledge, the multi-color changes for photovoltaic cell powered ECDs were rarely reported at present. Further, apart from deep blue color, PSCs powered MPV-based ECD can automatically switch between transparent, blue and magenta based on surrounding light intensities. These multiple colors of PSCs powered ECDs could contribute to evaluate the outside light intensity and energy storage status of EC capacitor.

Secondly, we agree with you that our reported energy storage capability is relatively weak from the discharging curve. However, the capacitance of DPV (6.7 mF/cm²) and MPV (7.1 mF/cm²) based EC capacitors were better than the reported EC capacitors (5.3 mF/cm², *Adv. Energy Mater.* **2017**, 7, 1602598; 5.3 mF/cm², *ACS Appl. Mater. Interfaces* **2017**, 9, 29872–29880), and two DPV based ECDs or MPV based ECDs can drive a red LED, which indicated an acceptable energy storage capability. Actually, as a proof of concept, our work did not emphasize on achieving high capacitance. Raising concentration of EC materials would benefit for improving the areal capacitance for gel-type ECD.

We agree that the faded colour in the device under a weak light illumination is a contradiction with energy storage capabilities. Generally the darkness of the colour is a reflection of the energy level of the ECD stored. The ECD would store more energy under a strong light illumination than under a weak light illumination. In the experiment we also found that “both charged ECDs with deep coloration (under strong light intensity) could drive a red LED successfully (Fig. 3c3-3c4) as supercapacitors” in manuscript. It is true the “memory effect” of colour change is usually accompanied with ion storage and energy storage, so most ECDs as supercapacitors have classic sandwich-type configuration with “memory effect”, we

utilize the “self-bleaching” feature of ECD to realize automatically light-modulating, which lead to a short-period energy storage capability, however, the ECD is integrated with PSC in our design, the energy harvested by the PSC will transfer to the ECD to provide a dynamic energy storage.

Finally, we are unable to agree with you about that “the solar cell and electrochromic device only show a comparable performance with the previous works”. Up to date, the stability performance of this DPV-based ECD was one of the best ones. Moreover, if we evaluate the optical contrast (ΔT) after 70000 cycles in reported investigations, this DPV-based ECD can achieve the best performance (ΔT remains to be 60.9%) in almost all reported viologen-based ECDs. We added a table (Table S1) to make comparisons with some reported excellent works in revised supporting information. From Table S1, it is obvious that our device demonstrated the best performance among the reported works so far.

Table S1. Performances of some viologen-based ECDs

EC materials	$\Delta T(\%)$	Stability	Ref
Monoheptyl-viologen	>80	3600 s without degradation (72 Cycles)	1
Diheptyl-viologen	>80	3600 s without degradation (~67 Cycles)	1
Vinyl benzyl viologen	65	60.5% remained after 10000 Cycles	2
DTFMBzV ^a	63.5	61.6% remained after 10000 Cycles	3
Nonyl viologen	55.2	53.8% remained after 10000 Cycles	4
PBT ^b	60	about 42% remained after 60000 Cycles	5
CPD ^c	63	53.7% remained after 40000 Cycles	6
DPV(this work)	74.3	60.9% remained after 70000 Cycles	-

Cycles: cycles; *a*: 1,1'-bis(3,5-bis(trifluoromethyl)-benzyl)-4,4'-bipyridine-1,1'-dium, *b*: 1,4-bis[(N-phosphono-2-ethyl)-4,4-bipyridinium]-methyl]-benzene tetrachloride, *c*: 1-(9-hexyl-9H-carbazole)-1-(propylphosphonicacid)-4,4-bipyridilium dichloride.

References

1. Kim JW, Myoung JM. Flexible and Transparent Electrochromic Displays with Simultaneously Implementable Subpixelated Ion Gel-Based Viologens by Multiple Patterning. *Adv. Funct. Mater.* **29**, 1808911 (2019).
2. Kao SY, Lu HC, Kung CW, Chen HW, Chang TH, Ho KC. Thermally Cured

- Dual Functional Viologen-Based All-in-One Electrochromic Devices with Panchromatic Modulation. *ACS Appl. Mater. Interfaces* **8**, 4175-4184 (2016).
3. Yu H-F, Chen K-I, Yeh M-H, Ho K-C. Effect of trifluoromethyl substituents in benzyl-based viologen on the electrochromic performance: Optical contrast and stability. *Sol. Energy Mater. Sol. Cells* **200**, 110020 (2019).
 4. Lu HC, Kao SY, Yu HF, Chang TH, Kung CW, Ho KC. Achieving Low-Energy Driven Viologens-Based Electrochromic Devices Utilizing Polymeric Ionic Liquids. *ACS Appl. Mater. Interfaces* **8**, 30351-30361 (2016).
 5. Weng D, Shi Y, Zheng J, Xu C. High performance black-to-transmissive electrochromic device with panchromatic absorption based on TiO₂-supported viologen and triphenylamine derivatives. *Org. Electron.* **34**, 139-145 (2016).
 6. Li M, Wei Y, Zheng J, Zhu D, Xu C. Highly contrasted and stable electrochromic device based on well-matched viologen and triphenylamine. *Org. Electron.* **15**, 428-434 (2014).

Reply to Reviewer 2

Comment 1: The position of Fig.3 (c1) should be adjusted, since the maximum value of x-axis is supposed to be 2000 instead of 200.

Answer: Thank you very much for pointing out this mistake, we adjusted Fig.3 and the label in c1 was correct in the revised manuscript. The figure was given below for your convenience.

Fig.3 UV-visible spectra of ECDs powered by two connected PSCs with decreased light density for DPV (a1) and MPV (b1); PSCs powered ECD in natural surrounding with strong light, weak light and no light (a2, a3, a4 for DPV-based ECD and b2, b3, b4 MPV-based ECD); Transmittance switching tests under strong light and no light of PSC-powered ECD at maximum absorption (c1: DPV, c2: MPV); Lighting LED powered by two connected ECDs (c3: DPV, c4: MPV).

Comment 2: Authors mentioned the ECDs showed an ultrahigh stability up to 70000

cycles. How did authors come to this statement comparable to other devices based on viologens? It is better to give a comparison table in the supplemental information.

Answer: Thank you very much for your kind suggestion. We supplemented a table in revised supporting information, which gave stability comparisons between DPV-based ECD and reported viologens-based ECDs (Table S1) to show the high stability of our devices. Thank you.

Table S1. Performances of some viologen-based ECDs

EC materials	$\Delta T(\%)$	Stability	Ref
Monoheptyl-viologen	>80	3600 s without degradation (72 Cyscs)	1
Diheptyl-viologen	>80	3600 s without degradation (~67 Cyscs)	1
Vinyl benzyl viologen	65	60.5% remained after 10000 Cyscs	2
DTFMBzV ^a	63.5	61.6% remained after 10000 Cyscs	3
Nonyl viologen	55.2	53.8% remained after 10000 Cyscs	4
PBT ^b	60	about 42% remained after 60000 Cyscs	5
CPD ^c	63	53.7% remained after 40000 Cyscs	6
DPV(this work)	74.3	60.9% remained after 70000 Cyscs	-

Cyscs: cycles; *a*: 1,1'-bis(3,5-bis(trifluoromethyl)-benzyl)-4,4'-bipyridine-1,1'-dium, *b*: 1,4-bis[[(N-phosphono-2-ethyl)-4,4-bipyridinium]-methyl]-benzene tetrachloride, *c*: 1-(9-hexyl-9H-carbazole)-1-(propylphosphonicacid)-4,4-bipyridilium dichloride.

Comment 3: What is the effect of the alkynyl group(s) on the performance of the ECD? Why is the di-pentynyl viologen-containing device much more stable than that of the mono-pentynyl viologen-containing one?

Answer: Thank you very much for pointing out your concern. The alkynyl groups serve as the electron-donor groups to stabilize the radical cations through the p- π interactions in two vertical directions. Compared to MPV with one pentynyl group, two alkynyl groups in same DPV molecule contribute to “lock” the molecule itself to reduce the probability of direct contact of intermolecular bipyridine moieties, which basically reduce the aggregation of radical cation species as shown in Fig. S2. The detailed discussions were given in supporting information

Fig. S2. Proposed interactions between alkyne groups and radical cations of bipyridine

Comment 4. For the NMR, the *J* should be italic *J*.

Answer: Thank you very much for your kind suggestion. We have corrected the mistake in revised manuscript.

Comment 5. In all the figures, the fonts should be united

Answer: Thank you very much for your kind suggestion. The fonts of all figures are united in revised manuscript.

Comment 6. 1-(pent-4-yn-1-yl)-[4, 4'-bipyridin]-1-ium chloride (MPV) and a di-alkynyl-substituted viologen of 1, 1'-di(pent-4-yn-1-yl)-[4,4'-bipyridine]-1,1'-diium dichloride (DPV) were synthesized and used for the fabrication of ECDs. Scheme 1 should be indicated here, not in the method part.

Answer: Thank you very much for your kind suggestion. Scheme 1 was moved to the introduction part of the revised manuscript.

Comment 7. GCD needs to be given after the first appearance of “Galvanostatic charge–discharge curves.” in the caption of Figure 2.

Answer: Thank you very much for your kind suggestion. *We have added the “GCD”*

after “Galvanostatic charge–discharge curves.” in the caption of Figure 2 in the revised manuscript.

Comment 8. “The detailed kinetic studies were developed in Supporting information” should be “The detailed kinetic studies were given in Supporting information”.

Answer: Thank you very much for your kind suggestion. We corrected the sentence of “The detailed kinetic studies were developed in Supporting information” to *“The detailed kinetic studies were given in Supporting information” in revised manuscript.*

Comment 9. “To our delight, 18.3% of PCE was achieved according to Fig. S6b, which was better than reported work 26, 27.” this sentence needs reword. The PCE of 18.3% is only a good reasonable one, since many devices with high PECs were reported.

Answer: Thank you very much for your kind suggestion. We rewrote the sentence in revised manuscript as follow: *“and reasonable high PCE of 18.3% was achieved according to Fig. S6b, which was better than some reported work 38, 39...”*

Comment 10. The discussion part is too short, I suggest authors move some discussions in the results part to this discussion section.

Answer: Thank you very much for your kind suggestion. We have moved some discussions in the results part to the discussion section in revised manuscript. I hope sincerely that the changes would meet your requirements.

Reply to Reviewer 3

Introduction section

Comment 1: it would be advisable to add a reference concerning the increase in global energy consumption.

Answer: Thank you for your kind suggestion. We have added a related reference in introduction part of revised manuscript (**Ref 1-3** in revised manuscript).

1. Cannavale A, Ayr U, Fiorito F, Martellotta F. Smart Electrochromic Windows to Enhance Building Energy Efficiency and Visual Comfort. *Energies* 13, 1449 (2020)
2. Xu J, Chen Y, Dai L. Efficiently photo-charging lithium-ion battery by perovskite solar cell. *Nat. Commun* 6, 8103 (2015).
3. Davy NC, et al. Pairing of near-ultraviolet solar cells with electrochromic windows for smart management of the solar spectrum. *Nat. Energy* 2, 17104(2017).

Comment 2: the authors should quote the fundamental works of S. Deb and C.G. Granqvist, about EC devices.

Answer: Thank you for your kind suggestion. We have cited the references describing fundamental works of S. Deb and C.G. Granqvist about EC devices in introduction part of revised manuscript (**Ref 5-7** in revised manuscript).

5. Deb SK. Reminiscences on the discovery of electrochromic phenomena in transition metal oxides. *Sol. Energy Mater. Sol. Cells* **39**, 191-201 (1995).
6. Granqvist CG. Oxide electrochromics: Why, how, and whither. *Sol. Energy Mater. Sol. Cells* **92**, 203-208 (2008).
7. Rui-Tao Wen, Claes-Göran Granqvist, Gunnar Niklasson: Eliminating Degradation and Uncovering Ion-trapping Dynamics in Electrochromic WO₃ Thin Films, *Nat. Mater.* **14**, 996-1001 (2015).

Comment 3: the authors of the reference [5] are not correctly cited in the text.

Answer: Thank you for your kind reminding. We have corrected this mistake in

revised manuscript. It is **Ref 11** in the revised manuscript.

11. Cannavale A, Eperon GE, Cossari P, Abate A, Snaith HJ, Gigli G. Perovskite photovoltaic cells for building integration. *Energy Environ. Sci.* **8**, 1578-1584 (2015).

Comment 4: the authors argue that the use of electricity to activate electrochromic windows is a deeply felt problem in the scientific field: in truth, as reported by several authors, the annual energy consumption of a commercial smart window is about 2.5 kWh/m², compared to the achievable energy saving, which is at least an order of magnitude higher.

Answer: Thank you for pointing out your concern. We agree with you about that the achievable energy saving of commercial smart window was much higher than its energy consumption in many cases. But generally a modern building has a large area of windows, the accumulation of this small amount of energy per m² is still a large number. We are aiming to achieve the goal of maximum energy saving through utilizing photovoltaic cells as the energy source for smart window.

For alleviating the argument, we rewrite related descriptions in revised manuscript.

(1) *In the abstract part: “but the requirement for external electrical supplies has significantly limited their critical applications” was revised to “but the requirement for the external electrical supplies can cause response-lag in smart windows or rearview mirrors.”*

(2) *In the introduction part: “the dependence on external power remains an inherent drawback for these windows” was revised to “Though remarkable energy savings were achieved, the dependence on external power caused response-lag in light modulation for ECDs.”*

Comment 5. the authors, on the other hand, do not refer to the benefits linked to visual comfort, connected to the modulation of the solar radiation obtainable;

Answer: Thank you for your kind suggestion. We have added the related descriptions in discussion part. (*“Smart modulation of the solar radiation contributes to offer a*

visual comfort for people. When the sunlight is strong, PSC harvests more energy and drive the EC into dark state, EC could permit less solar radiation to enter the building, therefore people inside will feel more comfortable under a weaker light than outside; on the other hand, when the sunlight is weak, PSC harvests less energy and drive the EC into relative bright state, EC could permit more solar radiation to enter the building, thus people inside will feel more comfortable under a brighter light” in the discussion part).

Comment 6. the authors declare that the so-called "memory effect" constitutes a limitation of common electrochromic glasses, but it is considered an asset for smart windows, worldwide.

Answer: Thank you for pointing out your concern. Indeed, smart windows with excellent “memory effect” can sustain the colored state for a long period without any output potential bias, and ECD can turn to transparent state if a reverse bias was applied. “Self-bleaching” refers to that ECD can turn to transparent state spontaneously from colored state in a short duration without any output potential bias. In our design, we just utilize the “self-bleaching” feature of ECD to realize automatically switching between colored and bleached states, which leads to appropriate light balance between outside and inside. For automatically adjusting transmittance, self-bleaching is more desirable in our design. *Thus, we deleted the sentence of “memory effect constitutes a limitation of common electrochromic glasses in the revised manuscript”. Thank you.*

Comment 7. however Figure S8 shows that the self-bleaching of the viologens is rather rapid and does not guarantee the typical "memory effect" requirements, also cited by the literature of the field (the recommended values of memory effect range from 2h to 12h).

Answer: Thank you for pointing out your concern. In reported self-powered characteristic ECDs, ECDs were easily converted to colored state under the power by

photovoltaic cells, but were hardly turn to bleached state because photovoltaic cells are unable to offer reverse potential. We did not aim to achieve excellent “memory effect” since we hope ECD could turn to transparent state spontaneously with the decrease of light intensity. By utilizing self-bleaching effect of all-in-one type ECD, our PSC-powered ECDs can alter their colors automatically in real time depending on the surrounding solar intensity.

Comment 8. With reference to the synthesis of viologens, the authors do not explain whether it was designed and implemented in this work for the first time or it is a protocol already appeared in literature;

Answer: Thank you for pointing out your concern. Mono-pentynyl viologen (MPV) is a novel compound, while di-pentynyl viologen (DPV) is not. We have stated the different originalities of two compounds in revised manuscript. (“...*two viologens including a novel mono-alkynyl-substituted viologen of 1-(pent-4-yn-1-yl)-[4,4'-bipyridin]-1-ium chloride (MPV) and a reported di-alkynyl-substituted viologen... in the introduction part*).

Comment 9. with reference to perovskite solar cells, some leading groups are not mentioned, which currently hold the records of photovoltaic conversion efficiency, also based on the annual NREL chart;

Answer: Thank you for your kind suggestion. We have cited the references of some leading groups and marked highest photovoltaic conversion efficiency in revised manuscript. (**Ref 22-30**). Since many groups are working on PSCs, some groups’ work may be still missed here. If we missed some groups’ work, please let us know again, and we can add in the reference part.

22. Mitzi DB. Introduction: Perovskites. *Chem. Rev.* **119**, 3033-3035 (2019).
23. Zheng X, *et al.* Managing grains and interfaces via ligand anchoring enables 22.3%-efficiency inverted perovskite solar cells. *Nat. Energy* **5**, 131-140 (2020).
24. Min H, *et al.* Efficient, stable solar cells by using inherent bandgap of α -phase formamidinium lead iodide. *Science* **366**, 749-753 (2019).

25. Lin YH, *et al.* A piperidinium salt stabilizes efficient metal-halide perovskite solar cells. *Science* **369**, 96-102 (2020).
26. Wolff CM, *et al.* Perfluorinated Self-Assembled Monolayers Enhance the Stability and Efficiency of Inverted Perovskite Solar Cells. *ACS Nano* **14**, 1445-1456 (2020).
27. Liu Z, *et al.* A holistic approach to interface stabilization for efficient perovskite solar modules with over 2,000-hour operational stability. *Nat. Energy* **5**, 596-604 (2020).
28. Zhao Y, *et al.* A Polymerization-Assisted Grain Growth Strategy for Efficient and Stable Perovskite Solar Cells. *Adv. Mater.* **32**, 1907769 (2020).
29. Zhu H, *et al.* Tailored Amphiphilic Molecular Mitigators for Stable Perovskite Solar Cells with 23.5% Efficiency. *Adv. Mater.* **32**, 1907757 (2020).
30. Jeong M, *et al.* Stable perovskite solar cells with efficiency exceeding 24.8% and 0.3-V voltage loss. *Science* **369**, 1615 (2020).

Results section

Comment 1. The authors report a careful examination of the stability of the viologens, even if the fast self-bleaching would cause - probably - an increase in the number of cycles required for coloring/bleaching, during operation, compared to the typical battery-like devices, based on transition metal oxides; would the rapid bleaching of electrochromic systems would require a continuous bias to the device, to keep it dark, as in SPDs systems? The authors could estimate the annual energy consumption, if they intend to propose this device for integration in buildings;

Answer: Thank you for pointing out your concern.

For all-in-one gel ECD itself, a continuous bias such as -1.6 V is needed to keep it deep dark. In our design, ECD is combined with PSC, the continuous bias provided by the PSC is enough to keep the ECD in an appropriate color state. ECD would keep colored state in varying degrees in the day time even if the sun was hidden by clouds thus, the complete cyclic behavior (switching between colored state and bleached state) would not be likely to happen.

The power to drive DPV based ECD to the deep colored state was 19.1 mJ/cm^2 according to Fig 2a3. If the smart window works 8 cycles per day, the annual energy consumption was calculated about 55.8 J/cm^2 ($0.16 \text{ kW}\cdot\text{h/m}^2$) per year approximately.

The power to sustain the deep colored state of DPV was 4.4 mW/cm^2 according to Fig 2a3. If the smart window works 8 h per day, the annual energy consumption was calculated $4.6 \times 10^4 \text{ J/cm}^2$ ($128.4 \text{ kW}\cdot\text{h/m}^2$) per year approximately.

The output power of PSC was approximately 18 mW/cm^2 according to Fig S6b. If the PSC works 8 h per day, the annual harvested energy was about $1.9 \times 10^5 \text{ J/cm}^2$ ($527.8 \text{ kW}\cdot\text{h/m}^2$).

Considering the power change of ECD in the whole day under a variable light intensity, the actual annual energy consumption or harvesting would be lower than estimated values for both ECD and PSC. Finally, the whole system can run smoothly without any extra energy supply. We have added the sentence of “*the annual energy consumption of the PSC powered ECD in detail in supporting information.*” in revised manuscript. Meanwhile, the description about the annual energy consumption was added in supporting information.

Comment 2. No reference is made to the stability of the perovskite-based solar cell, with inverted architecture, used in this device.

Answer: Thank you for your kind suggestion. We have cited the references about the stability of PSCs with inverted architecture in revised manuscript (Ref 37-38).

37. Chen Y, et al. Design of an Inorganic Mesoporous Hole-Transporting Layer for Highly Efficient and Stable Inverted Perovskite Solar Cells. *Adv. Mater.* 30, 1805660 (2018).

38. Jia L, Li B, Shang Y, Chen M, Wang G-W, Yang S. Double fullerene cathode buffer layers afford highly efficient and stable inverted planar perovskite solar cells. *Org. Electron.* 82, 105726 (2020).

Comment 3. Furthermore, it can be observed that this work announces the design and manufacture of a multifunction device, but only a juxtaposition of devices is presented in the text, mutually connected by an external electrical circuit.

Answer: Thank you for pointing out your concern. We admit that it is difficult to design and manufacture of a multifunctional device. In previous work (*Energy Environ. Sci.*, 2015, **8**, 1578—1584, *Mater. Horiz.*, 2016, **3**, 588—595), researchers also claimed their multifunctionality by combining the photovoltaic cells and ECD with external electrical circuit. Our work presented only a “proof of concept”, which was verified by juxtaposition of PSCs and ECDs. We are planning to integrate the ECD and PSC cells in one component (Fig. R1, which was given under the comment 4). We have carefully revised the description. *We deleted the sentence of “These studies indicated the PSC-powered ECDs have great potential applications in modern buildings and automobiles.” in page 15. We added the sentence of “...Herein, as a proof of concept, the gel type ECD with the all-in-one configuration was integrated with perovskite solar cell (PSC) for automatic light-adjustment...” in the introduction part.*

Comment 4. Other works of this type have appeared previously in the literature but they clearly declared to show just a "proof of concept", reporting at least a scheme of the device they intend to create. In fact, the difficulties of creating a multi-function device should not be overlooked, both in terms of layout and of processes: for example, what would the layout of the final device be? what is the relationship between the areas of the photovoltaic component and that of the electrochromic/capacitor component? is the photovoltaic component semi-transparent or opaque?

Answer: Thank you for pointing out your concern. We have added a figure (Fig.R1) to show the device we intend to create in the future as following.

Fig. R1 (figure for review only): Schematic diagram of the proposed final device

Fig. R1 show the proposed final device, the ECD served as the glass powered by connected PSCs which were inserted in window frame. The active area of single ECD and PSC device were 2 cm^2 and 0.115 cm^2 (with about 0.93 V of work voltage), and the ECD could be powered to coloured state well by two connected-PSCs (total areal: 0.23 cm^2). We have also noticed that large-area viologen-based all-in-one gel ECD could be driven by accepted voltage (-1.5 V, $14 \text{ cm} \times 12 \text{ cm}$, transparent-blue, *Electrochimica Acta*, 2019, 298, 533-540). Thus, large-area ECD may not have to require large area of PSCs if the voltage afforded by PSCs is high enough to drive the ECD in our future work. Finally, the photovoltaic component is opaque in our work.

Comment 5. Possibly, the authors should report some considerations about the potential toxicity of viologens used in this work.

Answer: Thank you very much for pointing out your concerns. Indeed, it is almost impossible to avoid the toxic and harmful issue when utilizing viologen derivatives. However, careful sealing would prevent DPV or MPV leaking out from fabricated ECDs with gel form. And as stable solid without any volatility at room temperature, DPV and MPV would not be absorbed by human bodies. Hence, strict handling of viologens and sealing of ECDs can largely minimize the toxicity of DPV or MPV to humans in this work.

Comment 6. Several typos are still visible in the current version of the manuscript: a thorough check would be useful.

Answer: Thank you for your kind reminding. We asked an English-native speaker to revise the manuscript carefully before resubmitting the revised manuscript.

REVIEWERS' COMMENTS

Reviewer #2 (Remarks to the Author):

The problems have been addressed. This manuscript can be accepted in the present form.

Reviewer #3 (Remarks to the Author):

In the first round of revisions, the authors improved the manuscript, based on the observations raised; in its current form, it can be considered suitable for publication in Nature Communications.

December 10, 2020

Dear reviewers,

We are grateful for your corrections and excellent suggestions for this work during the review process. Thank you very much for supporting our manuscript.

Thank you.

With best wishes.

Yanqing Tian, PhD., Professor
Materials Science and Engineering
Southern University of Science and Technology
Shenzhen, 518055, China
<http://www.researcherid.com/rid/C-1423-2013>

Answer to REVIEWERS' COMMENTS

Reviewer #2 (Remarks to the Author):

The problems have been addressed. This manuscript can be accepted in the present form.

Answer: Thank you for your agreement on the acceptance.

Reviewer #3 (Remarks to the Author):

In the first round of revisions, the authors improved the manuscript, based on the observations raised; in its current form, it can be considered suitable for publication in Nature Communications.

Answer: Thank you for your support.